# National trends and socioeconomic inequalities in the Composite Index of Severe Anthropometric Failure among children under five in Bangladesh

Md Fuad Al Fidah[1,2]*, Md Nafis Fuad[3], Tahia Tul Islam[3], Tasnuva Sarowar[4], Syeda Sumaiya Efa[1]

1 National Institute and Preventive and Social Medicine, Dhaka, Bangladesh, 2 NRD, icddr,b, Dhaka, Bangladesh, 3 Research Wing, Bibliophile, Dhaka, Bangladesh, 4 Department of Oncological Sciences, Icahn School of Medicine at Mount Sinai, New York, United States of America

* fuadml@gmail.com

## Abstract

Severe undernutrition in early childhood remains a major public health challenge in Bangladesh. The Composite Index of Severe Anthropometric Failure (CISAF) provides a more comprehensive metric of undernutrition. The study examined the prevalence, trends, and socioeconomic disparities in CISAF among children under five in Bangladesh using three nationally representative surveys. Data were extracted from the Bangladesh Demographic and Health Survey (BDHS) 2017–18, Multiple Indicator Cluster Survey (MICS) 2019, and BDHS 2022. The study scope was restricted to the most recent survey rounds to reflect current conditions and to maintain consistency with the programme period now used in national planning. Descriptive, regression, and inequality analyses (slope index of inequality [SII], relative index of inequality [RII]) were performed to assess trends and predictors. The Approximate test of homogeneity of odds ratios was used to examine the homogeneity of odds across survey years. Among 33452 children under five, CISAF prevalence significantly declined from 10.3% in 2017–18 to 10.1% in 2019 and finally, 7.6% in 2022. Higher odds of CISAF were observed among older age groups compared with younger groups across all the surveys (aOR: 1.03, 1.10 and 1.02, respectively). However, lower odds were linked to wealthier families and higher education of the mother. SII declined from -0.12 (2017–18) to -0.09 (2022), while RII ranged from 1.41 to 1.54, indicating persistent inequality despite national improvements in severe undernutrition. Severe undernutrition among children under five has declined, but clear socioeconomic gaps remain. Children from poorer households and those with less educated mothers continue to face higher levels of severe anthropometric failure. Equity-focused approaches, including targeted community services, social protection support, and wider access to maternal health information, may help narrow these gaps. Integrating CISAF into routine surveillance could improve identification of high-risk groups and support stronger programme planning.

**Data availability statement:** Data is available in a public, open-access website at https://mics.unicef.org/surveys and https://dhsprogram.com/.

**Funding:** The authors received no specific funding for this work.

**Competing interests:** The authors have declared that no competing interests exist.

## Introduction

Child undernutrition remains a significant public health concern in low- and middle-income countries (LMICs), especially in South Asia [1]. Bangladesh, despite remarkable progress in maternal and child health, continues to face high levels of undernutrition among children under the age of five [1] Undernutrition in early childhood is associated with long-term consequences, inclu.ding poor physical growth, impaired cognitive development, reduced productivity in adulthood, and increased risk of morbidity and mortality [2,3]. Moreover, undernutrition in childhood can affect the next generation. A girl who was undernourished in childhood may have problems giving birth to a healthy baby in future [1]. This is partly due to the persistence of nutritional deprivation across the life course, where undernourished children are more likely to remain malnourished into adulthood. Globally, child undernutrition remains a leading driver of morbidity and mortality. Recent estimates report 149 million children under five stunted and 45 million wasted, underscoring the ongoing scale of the problem in low- and middle-income countries, including South Asia [4–6]. These burdens concentrate among socioeconomically disadvantaged groups and settings with limited WASH and health-service access.

Given the serious consequences of childhood undernutrition, governments have committed to global targets aimed at reducing chronic undernutrition (stunting) by 40% by 2025 and lowering the prevalence of acute undernutrition (wasting) to below 5% among children under five years of age [7]. These commitments align with Sustainable Development Goal (SDG) 2 [3]. Moreover, adequate nutrition is considered essential for achieving several other SDGs [8].

Traditionally, the nutritional status of children has been assessed using three indicators: stunting (low height-for-age), wasting (low weight-for-height), and underweight (low weight-for-age) [9]. Conventional indicators such as stunting, wasting, and underweight capture different aspects of child malnutrition but cannot estimate the total burden at the population level [5,9,10]. To overcome this, Svedberg proposed the Composite Index of Anthropometric Failure (CIAF), which incorporates all forms of failure [1,11,12]. Later, Vollmer et al. (2017) introduced the Composite Index of Severe Anthropometric Failure (CISAF), providing a more refined tool for assessing severe undernutrition in resource-poor settings [3]. This index identifies a more vulnerable population than the CIAF or single malnutrition indicators.

CISAF is particularly useful for policy and planning purposes, as it allows identification of children experiencing the most critical forms of nutritional deprivation. Understanding how severe undernutrition is distributed across different population groups and how it evolves over time is essential for designing targeted interventions. Studies using CISAF are still limited, especially in the South Asian context. One study in Bangladesh has applied CISAF to examine urban-rural disparities in severe undernutrition and found significant inequalities based on socioeconomic and demographic characteristics [1]. Although few examined the prevalence and factors associated with CISAF in Bangladesh [1,3], studies have tracked national trends in severe anthropometric failure while simultaneously quantifying socioeconomic inequalities using CISAF. Using three recent nationally representative surveys, i.e.,

the Bangladesh Demographic and Health Survey (BDHS) 2017–18, the Multiple Indicator Cluster Survey (MICS) 2019, and the BDHS 2022, this study aimed to examine national trends and socioeconomic inequalities in the CISAF among children under five in Bangladesh.

## Methods

### Study design and data sources

Data were drawn from three nationally representative surveys: BDHS 2017–18, MICS 2019, and BDHS 2022. Both BDHS and MICS employed a two-stage stratified cluster sampling design. Enumeration areas were first selected with probability proportional to size, followed by systematic sampling of households. All ever-married women aged 15–49 years were interviewed, and anthropometric measurements were collected for their children under five.

The BDHS surveys employed a two-stage stratified sampling design. In the first stage, enumeration areas (EAs), also known as primary sampling units (PSUs), were selected using probability proportional to size, based on the 2011 Population and Housing Census conducted by the Bangladesh Bureau of Statistics (BBS) [13,14]. In the second stage, a fixed number of households were chosen from each PSU using systematic random sampling. In BDHS 2017–18, anthropometric measurements were available for 8,759 children under five. In BDHS 2022, although 8,784 children were eligible, only 4,420 had complete anthropometric information due to missing values for height, weight, or age [13,14].

The MICS 2019 survey followed a similar two-stage stratified cluster sampling approach, covering all administrative divisions of Bangladesh. The survey was conducted by the Bangladesh Bureau of Statistics with technical and financial support from the UNICEF [15]. In total, 24686 children under five years were eligible for interview. Anthropometric data in all three surveys were collected by trained personnel using calibrated equipment and standardised procedures. All three datasets were harmonised to ensure comparability of variables. Details of the survey methodology, sample design, and data collection tools are available in the respective national survey reports [13–15].

*Sample selection.* Fig 1 briefly shows the procedure used to select participants for this analysis. Children aged 0–59 months were included from the publicly available datasets and were merged with the women's dataset (wm) to obtain additional background variables. Cases with missing information on age, anthropometry or with values aligning with WHO guidelines for defining biologically implausible anthropometric values were excluded [9]. Biologically implausible anthropometric values were excluded using the WHO flag variables provided within each survey dataset. These survey-generated flags follow the WHO 2006 Growth Standards and identify extreme Z-scores outside accepted ranges. Implausible values were defined as weight-for-height Z-score (WHZ) <-5 or >5 SD, height-for-age Z-score (HAZ) <-6 or >6 SD, and weight-for-age Z-score (WAZ) <-6 or >5 SD [16,17].

After excluding missing and implausible values, the final analytic sample comprised 33452 children under five. From BDHS 2017–18, 7773 children were included, 21609 from MICS 2019, and 4070 from BDHS 2022.

*Study variables.* *Outcome variable: Composite index of severe anthropometric failures (CISAF);* In this analysis, the CISAF among children under five was considered the outcome variable. A child was defined as severely stunted, severely wasted, or severely underweight if their Z-score for length/height-for-age (HAZ/LAZ), weight-for-height/length (WHZ/WLZ), or weight-for-age (WAZ), respectively, was less than -3 standard deviations (SD) from the WHO reference population median [18]. Severe nutritional status among children under five was classified into seven mutually exclusive categories: (A) no severe failure; (B) severe wasting only; (C) severe wasting with severe underweight; (D) severe wasting with severe stunting and severe underweight; (E) severe stunting with severe underweight; (F) severe stunting only; and (G) severe underweight only (S1 Fig). Children were considered severely undernourished if they fell into any category from B to G.

*Independent variables:* The selection of independent variables was informed by previously identified socioeconomic determinants reported in the literature, as well as availability across all the survey years [1,3,13–15]. These variables included Child's age (in months), mother's age (in years), father's age (in years), antenatal care (in complete numbers),

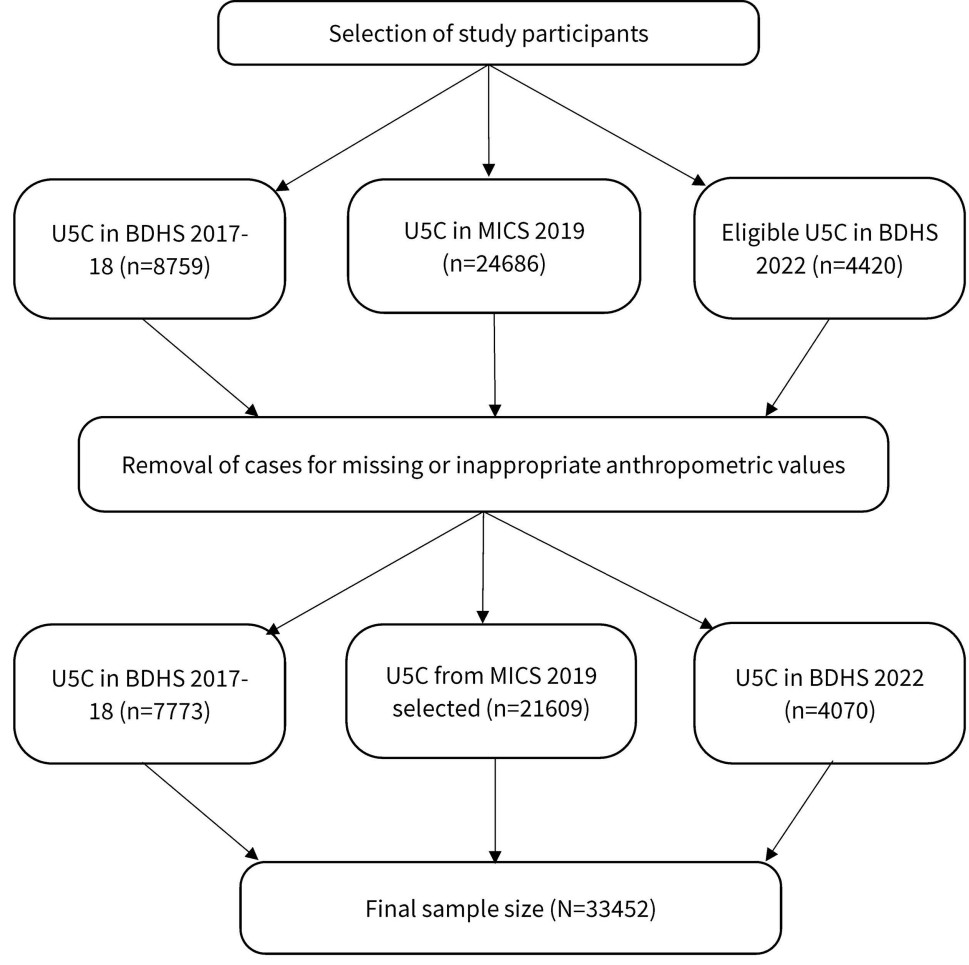

BDHS: Bangladesh Demographic and Health Survey; MICS: Multiple Indicator Cluster Survey

**Fig 1. Steps of selection of children under five from three nationally representative surveys.**

sex of the child (male/female), residence (rural/urban), diarrhoea (yes/no), fever (yes/no), cough (yes/no), wealth index quintile (poorest/poor/middle/rich/richest), mother's education (pre-primary or none/primary/secondary/higher), received any prenatal care (no/yes) [19], and maternal media exposure (not exposed/exposed).

*Operational definitions for independent variables. Diarrhoea:* Mothers or primary caregivers were asked whether the child had diarrhoea during the two weeks before the survey. Responses were recorded as yes or no.

*Fever:* Mothers or caregivers reported whether the child experienced fever in the two weeks before the survey. Responses were categorised as yes or no.

*Cough:* Mothers or caregivers reported if the child had a cough in the two weeks before the survey. Responses were categorised as yes or no.

***Wealth index quintile.*** The wealth index functions as a proxy for household economic status, based on the premise that material assets, housing characteristics, and access to basic services collectively reflect a household's relative position within the socioeconomic hierarchy [20]. Instead of relying on direct measures of income or expenditure, which

are often prone to reporting inaccuracies and data collection challenges, the index draws on observable indicators such as ownership of durable goods, quality of housing, and access to water and sanitation facilities. Respondents were subsequently classified into five categories: poorest, poor, middle, rich, and richest. This asset-based approach provides a reliable and pragmatic means of estimating relative economic standing within large-scale demographic surveys [20].

*Maternal media exposure.* Mothers' media exposure was calculated using three items: 'frequency of reading newspaper or magazine', 'frequency of listening to the radio' and 'frequency of watching TV'. The variable had two categories, namely not exposed (0 score) and exposed [9,21].

*Statistical analysis.* Data were analysed using STATA 19 (StataCorp, College Station, Texas, USA). Categorical variables were presented using frequency and percentage. For continuous variables, median and interquartile range (IQR) were reported. The prevalence of the CISAF was estimated for each survey round and across key background characteristics. The association between CISAF and background characteristics was explored using Pearson's Chi-square test for categorical variables. For continuous variables, the Mann-Whitney U test was applied.

*Trend test for CISAF across survey years.* To assess trends in CISAF prevalence over time, the Cochran-Armitage test for trend was used, which evaluates whether there is a statistically significant monotonic trend across ordered groups (i.e., survey years). This test is particularly appropriate for binary outcomes over ordered time points.

*Homogeneity of odds ratios.* To complement this, Mantel-Haenszel (MH) stratified analyses were used to compute stratum-adjusted pooled odds ratios for unadjusted associations between each independent variable and CISAF across survey years. The MH method allowed assessment of overall associations while adjusting for survey year as a stratification factor. Approximate tests of homogeneity of odds ratios (HOR) were conducted to evaluate the homogeneity of odds ratios across years. A significant HOR test suggests that the association between a predictor and CISAF varies significantly across survey periods. For ordinal variables (e.g., wealth index, maternal education), the MH procedure included a score test for trend in odds.

*Inequality index.* To assess socioeconomic inequalities in severe undernutrition, both the Slope Index of Inequality (SII) and the Relative Index of Inequality (RII) were estimated across household wealth quintiles. These measures were chosen because they account for the entire distribution of the socioeconomic variable rather than comparing only the extreme groups. For each survey year, children were ranked according to the cumulative distribution of the DHS wealth index, which served as a continuous relative rank variable ranging from 0 (least advantaged) to 1 (most advantaged) [9,22]. Logistic regression models were fitted with this rank variable as the independent predictor and the CISAF as the binary outcome [9].

The SII represents the absolute difference in the predicted prevalence of CISAF between children at the lowest end (rank = 0) and the highest end (rank = 1) of the wealth distribution. Thus, it quantifies the gap in severe undernutrition between the poorest and richest children while incorporating information from all intermediate groups. A value of zero indicates no absolute inequality [9,23].

The RII expresses relative inequality by dividing the predicted prevalence at the lowest rank by that at the highest rank, providing the proportional difference in CISAF between the poorest and richest children. An RII of one indicates no relative inequality [9,23].

Sex of the child was included as a covariate to adjust for possible differences in nutritional risk by sex. The study's inequality analyses (SII and RII) focus on socioeconomic axes (household wealth); sex is included to control confounding rather than to define the inequality measures.

*Multiple binomial logistic regression.* To identify determinants of severe undernutrition, binomial logistic regression models were fitted separately for each survey round, with the CISAF as the dependent variable. All selected independent variables were included in the models simultaneously, adjusting for each other. Results were presented as adjusted odds ratios (AORs) with 95% confidence intervals (CIs). Multi-collinearity among covariates was assessed using the variance inflation factor (VIF), with a threshold value of 10 indicating potential concern.

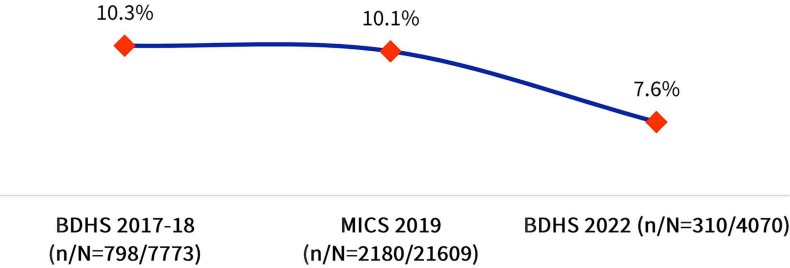

**Ethics statement.** The current study used data from a publicly available website. As secondary analysis was conducted, no ethical clearance was required. All original data collection received ethical clearance from the relevant national review boards. Verbal informed consent was taken from each of the participants prior to data collection.

## Results

Data from a total of 33452 children under five were analyse for the study. The steps of the selection of participants are presented in Fig 1.

Fig 2 illustrates the national trend in the prevalence of CISAF among children under five in Bangladesh across three nationally representative surveys. The overall prevalence of CISAF showed a declining trend over time. In 2017–18, the prevalence was 10.3%. This figure remained nearly unchanged in 2019, recorded at 10.1%. However, a notable decline was observed in 2022, where the prevalence dropped to 7.6% and this downward trend was significant ($p < 0.001$). A test for departure from linearity was also significant ($p = 0.001$), suggesting that while a downward trend is present, the decline may not have followed a strictly linear pattern (Fig 2).

S2 Fig shows the distribution of CISAF categories from 2017-18–2022. The share of children with no severe failure increased from 89.7% to 92.4%, indicating overall improvement. Among the failure types, severe stunting only remained the most common, though it declined from 5.95% to 3.17%. Other single severe failures, such as severe wasting only and severe underweight only, remained below 1% across all years. Combined failures, like severe stunting and severe underweight, declined slightly (2.02% to 1.62%), while all forms of severe undernutrition remained rare (0.12%, 0.21% and 0.17% respectively) (S2 Fig).

In the 2017–18 survey, antenatal care, residence, wealth status, mother's education, prenatal care, and maternal media exposure were significantly associated with CISAF prevalence ($p < 0.05$). While in the 2019 survey, wealth index, mother's education, diarrhoea incidence, prenatal care, and media exposure remained significantly linked to severe undernutrition. Finally, in the 2022 data, a significant association was observed between a child's age, wealth, mother's education, antenatal care, and media exposure and CISAF (Table 1).

Across all three survey years, a child's increasing age was consistently associated with higher odds of CISAF, with the highest odds observed in 2017–18 (aOR: 1.03; 95% CI: 1.02-1.04). The household wealth index also demonstrated a significant association over time. Compared to the poorest children, those from middle to richest households were linked with progressively lower odds of CISAF. For instance, in 2017–18, children from the richest households were associated with 0.56 (95% CI: 0.38-0.84) times lower odds, with a similar association observed in 2019 and 2022. Maternal education

10.3%   10.1%

7.6%

BDHS 2017-18 (n/N=798/7773)   MICS 2019 (n/N=2180/21609)   BDHS 2022 (n/N=310/4070)

*p-value for: Cochran-Armitage test=< 0.001, test of departure from trend=0.001*
*BDHS: Bangladesh Demographic and Health Survey; MICS: Multiple Indicator Cluster Survey*

**Fig 2. Trend of presence of CISAF from 2017-18 to 2022.**

**Table 1. Association between independent and dependent variables.**

**BDHS 2017–18**

| Variables | Total n (%) | CISAF | | p-value |
|---|---|---|---|---|
| | | No, n (%) | Yes, n (%) | |
| **Child's age in months** | 28 (13, 44) | 28 (13, 44) | 28 (17, 41) | 0.620 |
| **Mother's age in years** | 25 (21, 29) | 25 (21, 29) | 25 (21, 30) | 0.540 |
| **Father's age in years** (n = 7646) | 33 (29, 38) | 33 (29, 38) | 32 (28, 38) | 0.790 |
| **Antenatal care** (n = 4585) | 3 (2, 6) | 3 (2, 6) | 3 (1, 5) | <0.001 |
| **Sex** (male), n (%) | 4048 (52.1) | 3623 (51.9) | 425 (53.3) | 0.480 |
| **Residence** (urban), n (%) | 5125 (65.9) | 4570 (65.5) | 555 (69.5) | 0.023 |
| **Diarrhoea** (yes), n (=7770) (%) | 389 (5.0) | 353 (5.1) | 36 (4.5) | 0.500 |
| **Fever** (yes), n (=7769) (%) | 5163 (66.5) | 4653 (66.7) | 510 (64.0) | 0.120 |
| **Cough** (yes), n (%) | 2606 (33.5) | 2319 (33.3) | 287 (36.0) | 0.680 |
| **Wealth index quintile** | 1733 (22.3) | | | <0.001 |
| Poorest | 1568 (20.2) | 1471 (21.1) | 262 (32.8) | |
| Poor | 1410 (18.1) | 1373 (19.7) | 195 (24.4) | |
| Middle | 1546 (19.9) | 1279 (18.3) | 131 (16.4) | |
| Rich | 1516 (19.5) | 1419 (20.3) | 127 (15.9) | |
| Richest | 1733 (22.3) | 1433 (20.5) | 83 (10.4) | |
| **Mother's education** | | | | <0.001 |
| Pre-primary or none | 555 (7.1) | 450 (6.5) | 105 (13.2) | |
| Primary | 2247 (28.9) | 1935 (27.7) | 312 (39.1) | |
| Secondary | 3675 (47.3) | 3347 (48.0) | 328 (41.1) | |
| Higher | 1296 (16.7) | 1243 (17.8) | 53 (6.6) | |
| **Received prenatal care** (yes), n (=4585) (%) | 4214 (91.9) | 3787 (92.5) | 427 (87.0) | <0.001 |
| **Maternal media exposure** (exposed), n (%) | 4956 (63.8) | 2434 (34.9) | 383 (48.0) | <0.001 |

**MICS 2019**

| Variables | Total n (%) | CISAF | | p-value |
|---|---|---|---|---|
| | | No, n (%) | Yes, n (%) | |
| **Child's age in months** | 30 (15, 44) | 30 (15, 45) | 29 (16, 42) | 0.390 |
| **Mother's age in years** (n = 21543) | 26 (23, 31) | 26 (23, 31) | 26 (22, 31) | 0.550 |
| **Father's age in years** (n = 21543) | 34 (30, 39) | 34 (30, 39) | 34 (29, 39) | 0.013 |
| **Antenatal care** (n = 8380) | 3 (2, 5) | 3 (2, 5) | 3 (2, 5) | 0.860 |
| **Sex** (male), n (%) | 11147 (51.6) | 10002 (51.5) | 1145 (52.5) | 0.360 |
| **Residence** (urban), n (%) | 17647 (81.7) | 3602 (18.5) | 360 (16.5) | 0.020 |
| **Diarrhoea** (yes), n (=21596) (%) | 1503 (7.0) | 1317 (6.8) | 186 (8.5) | 0.002 |
| **Fever** (yes), n (=21601) (%) | 16579 (76.8) | 4483 (23.1) | 539 (24.7) | 0.081 |
| **Cough** (yes), n (=21601) (%) | 5022 (23.2) | 4162 (21.4) | 504 (23.1) | 0.067 |
| **Wealth index quintile** | | | | <0.001 |
| Poorest | 5305 (24.5) | 4562 (23.5) | 743 (34.1) | |
| Poor | 4520 (20.9) | 4018 (20.7) | 502 (23.0) | |
| Middle | 4124 (19.1) | 3751 (19.3) | 373 (17.1) | |
| Rich | 4088 (18.9) | 3778 (19.4) | 310 (14.2) | |
| Richest | 3572 (16.5) | 3320 (17.1) | 252 (11.6) | |

*(Continued)*

**Table 1.** (Continued)

**BDHS 2017–18**

| Variables | Total n (%) | CISAF | | p-value |
|---|---|---|---|---|
| | | No, n (%) | Yes, n (%) | |
| **Mother's education** | | | | <0.001 |
| Pre-primary or none | 2272 (10.5) | 1931 (9.9) | 341 (15.6) | |
| Primary | 5209 (24.1) | 4543 (23.4) | 666 (30.6) | |
| Secondary | 10757 (49.8) | 9804 (50.5) | 953 (43.7) | |
| Higher | 3371 (15.6) | 3151 (16.2) | 220 (10.1) | |
| **Received prenatal care** (yes), n (=10387) (%) | 8386 (80.7) | 7601 (81.6) | 785 (73.6) | <0.001 |
| **Maternal media exposure** (exposed), n (%) | 2001 (19.3) | 12368 (63.7) | 1191 (54.6) | <0.001 |

**BDHS 2022**

| Variables | Total n (%) | CISAF | | p-value |
|---|---|---|---|---|
| | | No, n (%) | Yes, n (%) | |
| **Child's age in months** | 29 (13, 45) | 29 (13, 44) | 32 (20, 47) | 0.002 |
| **Mother's age in years** | 26 (22, 30) | 26 (22, 30) | 26 (22, 31) | 0.310 |
| **Father's age in years** (n = 2355) | 35 (30, 39) | 35 (30, 39) | 35 (30, 40) | 0.610 |
| **Antenatal care** (n = 2355) | 3 (2, 4) | 3 (2, 5) | 3 (1, 4) | 0.001 |
| **Sex** (male), n (%) | 2090 (51.4) | 1935 (51.5) | 155 (50.0) | 0.620 |
| **Residence** (urban), n (%) | 2770 (68.1) | 1212 (32.2) | 88 (28.4) | 0.160 |
| **Diarrhoea** (yes), n (=4065) (%) | 207 (5.1) | 193 (5.1) | 14 (4.5) | 0.640 |
| **Fever** (yes), n (=4063) (%) | 2775 (68.3) | 1192 (31.7) | 96 (31.2) | 0.830 |
| **Cough** (yes), n (=4062) (%) | 1288 (31.7) | 1071 (28.5) | 81 (26.2) | 0.380 |
| **Wealth index quintile** | | | | <0.001 |
| Poorest | 874 (21.5) | 768 (20.4) | 106 (34.2) | |
| Poor | 802 (19.7) | 739 (19.7) | 63 (20.3) | |
| Middle | 817 (20.1) | 753 (20.0) | 64 (20.6) | |
| Rich | 787 (19.3) | 742 (19.7) | 45 (14.5) | |
| Richest | 790 (19.4) | 758 (20.2) | 32 (10.3) | |
| **Mother's education** | | | | <0.001 |
| Pre-primary or none | 248 (6.1) | 205 (5.5) | 43 (13.9) | |
| Primary | 934 (22.9) | 834 (22.2) | 100 (32.3) | |
| Secondary | 2120 (52.1) | 1992 (53.0) | 128 (41.3) | |
| Higher | 768 (18.9) | 729 (19.4) | 39 (12.6) | |
| **Received prenatal care** (yes), n (=2355) (%) | 2180 (92.6) | 2035 (92.8) | 145 (89.5) | 0.120 |
| **Maternal media exposure** (exposed), n (%) | 175 (7.4) | 2139 (56.9) | 142 (45.8) | <0.001 |

CISAF: Composite index of severe anthropometric failure; BDHS: Bangladesh Demographic and Health Survey; MICS: Multiple Indicator Cluster Survey.

remained a strong factor across surveys. Children of mothers with secondary or higher education were consistently associated with significantly lower odds of severe undernutrition. In 2017–18, secondary education was associated with a 0.59 (95% CI: 0.41-0.86) times lower odds, and higher education with 0.35 times (95% CI: 0.21-0.57) lower odds. Similar patterns were found in subsequent years (Table 2).

Table 3 shows the stratum-adjusted pulled odds ratios (ORs) and homogeneity tests for different background characteristics and their association with CISAF, using the MH method across three survey years. The HOR test was used to check if the OR was similar across years. Child's age showed no overall association with CISAF (p = 0.597), but the HOR test was significant

**Table 2. Binomial logistic regression analysis stratified by survey year.**

| Variables | BDHS 2017–18 (n = 4526) | | MICS 2019 | | BDHS 2022 | |
|---|---|---|---|---|---|---|
| | aOR (95% CI) | p-value | aOR (95% CI) | p-value | aOR (95% CI) | p-value |
| **Child's age in months** | 1.03 (1.02-1.04) | <0.001 | 1.01 (1.00-1.01) | 0.025 | 1.02 (1.01-1.04) | 0.009 |
| **Mother's age in years** | 0.99 (0.96-1.02) | 0.505 | 1.00 (0.98-1.02) | 0.949 | 1.05 (1.00-1.09) | 0.054 |
| **Father's age in years** | 0.99 (0.97-1.01) | 0.420 | 1.00 (0.98-1.01) | 0.805 | 0.96 (0.93-1.00) | 0.055 |
| **Antenatal care** | 0.96 (0.92-1.00) | 0.055 | 1.04 (1.00-1.08) | 0.056 | 0.97 (0.89-1.07) | 0.550 |
| **Sex** | Reference: Female | | | | | |
| Male | 1.20 (0.99-1.46) | 0.062 | 1.14 (0.98-1.32) | 0.095 | 1.36 (0.97-1.90) | 0.075 |
| **Residence** | Reference: Rural | | | | | |
| Urban | 1.20 (0.95-1.51) | 0.119 | 1.07 (0.87-1.31) | 0.523 | 0.92 (0.62-1.37) | 0.688 |
| **Diarrhoea** | Reference: No | | | | | |
| Yes | 0.90 (0.61-1.33) | 0.597 | 1.22 (0.95-1.57) | 0.121 | 0.66 (0.30-1.45) | 0.300 |
| **Fever** | Reference: No | | | | | |
| Yes | 1.10 (0.87-1.39) | 0.431 | 0.90 (0.74-1.09) | 0.275 | 0.89 (0.58-1.35) | 0.575 |
| **Cough** | Reference: No | | | | | |
| Yes | 0.92 (0.73-1.16) | 0.479 | 1.05 (0.87-1.27) | 0.608 | 0.92 (0.60-1.42) | 0.702 |
| **Wealth index quintile** | Reference: Poorest | | | | | |
| Poor | 0.85 (0.65-1.12) | 0.256 | 0.81 (0.65-1.02) | 0.078 | 0.82 (0.51-1.34) | 0.433 |
| Middle | 0.68 (0.49-0.93) | 0.015 | 0.76 (0.60-0.97) | 0.028 | 0.92 (0.56-1.52) | 0.751 |
| Rich | 0.57 (0.41-0.80) | 0.001 | 0.65 (0.50-0.85) | 0.001 | 0.75 (0.43-1.31) | 0.317 |
| Richest | 0.56 (0.38-0.84) | 0.005 | 0.66 (0.49-0.89) | 0.006 | 0.48 (0.24-0.97) | 0.042 |
| **Mother's education** | Reference: Pre-primary or none | | | | | |
| Primary | 0.78 (0.54-1.11) | 0.164 | 0.74 (0.56-0.99) | 0.044 | 0.58 (0.32-1.05) | 0.073 |
| Secondary | 0.59 (0.41-0.86) | 0.005 | 0.64 (0.48-0.84) | <0.001 | 0.35 (0.19-0.64) | <0.001 |
| Higher | 0.35 (0.21-0.57) | <0.001 | 0.48 (0.34-0.67) | <0.001 | 0.56 (0.28-1.14) | 0.112 |
| **Received prenatal care** | Reference: No | | | | | |
| Yes | 0.87 (0.62-1.21) | 0.406 | — | — | 1.21 (0.64-2.30) | 0.560 |
| **Maternal media exposure** | Reference: Not exposed | | | | | |
| Exposed | 0.92 (0.73-1.15) | 0.465 | 1.03 (0.87-1.23) | 0.707 | 0.86 (0.60-1.24) | 0.422 |

—: Omitted due to collinearity; CISAF: Composite index of severe anthropometric failure; aOR: Adjusted Odds ratio; CI: Confidence interval; BDHS: Bangladesh Demographic and Health Survey; MICS: Multiple Indicator Cluster Survey; Reference category for CISAF: No.

(p = 0.006), indicating that the relationship varied across years. In 2022, older children had slightly higher odds of CISAF (OR = 1.01), while in earlier years the effect was not significant. Antenatal care visits were associated with lower odds in the pooled model (OR = 0.95; 95% CI: 0.93-0.97; p < 0.001), and the HOR test was also significant (p = 0.002), suggesting that the strength of this effect changed over time. In both 2017–18 and 2022, ANC was significant, while for 2019 it was not (Table 3).

Household wealth (OR: 0.80; 95% CI: 0.78-0.82) and maternal education (OR: 0.67; 95% CI: 0.64-0.70) showed strong and consistent negative associations with CISAF across all years. This indicated that belonging to a higher category showed lower observed odds of CISAF. However, the HOR test for maternal education was highly significant (p < 0.001), showing that its protective effect varied between years (Table 3).

Postnatal care (OR: 0.61; 95% CI: 0.54-0.70) and maternal media exposure (OR: 0.66; 95% CI: 0.61-0.71) were also protective across all survey rounds, and no significant heterogeneity across years. Diarrhoea (OR: 1.17; 95% CI:1.02-1.35) and fever (OR: 1.09; 95% CI: 1.01-1.18) had modest associations in the pooled analysis (OR = 1.17 and 1.09, respectively), but only diarrhoea showed potential variation across years (p = 0.093) (Table 3).

**Table 3. Pooled Odds Ratios and Homogeneity Tests for Predictors of CISAF Across Survey Years in Bangladesh.**

| Variable | Pooled OR (95% CI) | Year-wise pooled OR | | | MH p-value | HOR p-value |
|---|---|---|---|---|---|---|
| | | BDHS 2017–18 | MICS 2019 | BDHS 2022 | | |
| Child's age (in months) | 1.00 (0.99-1.00) | 1.00 | 0.99 | 1.01* | 0.597 | 0.006 |
| Mother's age (in years) | 1.00 (0.99-1.01) | 0.99 | 0.99 | 1.02 | 0.821 | 0.293 |
| Father's age (in years) | 0.99 (0.99-1.00) | 1.00 | 0.99* | 1.01 | 0.184 | 0.125 |
| ANC visits (in numbers) | 0.95 (0.93-0.97) | 0.92* | 1.00 | 0.92* | <0.001 | 0.002 |
| Sex (male) | 1.03 (0.96-1.11) | 1.05 | 1.04 | 0.94 | 0.345 | 0.703 |
| Residence (urban) | 0.85 (0.78-0.93) | 0.83* | 0.87* | 0.83 | <0.001 | 0.895 |
| Diarrhoea (yes), n (=4065) | 1.17 (1.02-1.35) | 0.89 | 1.28* | 0.88 | 0.030 | 0.093 |
| Fever (yes) | 1.09 (1.01-1.18) | 1.13 | 1.10 | 0.97 | 0.034 | 0.609 |
| Cough (yes) | 1.05 (0.98-1.15) | 1.03 | 1.10 | 0.89 | 0.175 | 0.305 |
| Wealth index | 0.80 (0.78-0.82) | 0.77* | 0.82* | 0.77* | <0.001 | 0.091 |
| Mother's education | 0.67 (0.64-0.70) | 0.59* | 0.71* | 0.59* | <0.001 | <0.001 |
| Received prenatal care (yes) | 0.61 (0.54-0.70) | 0.54* | 0.63* | 0.66 | <0.001 | 0.626 |
| Maternal media exposure (exposed) | 0.66 (0.61-0.71) | 0.58* | 0.69* | 0.64* | <0.001 | 0.157 |

OR: Odds ratio; CI: Confidence interval; MH: Mantel–Haenszel; HOR: Approximate test of homogeneity of odds ratios *: p-value<0.05; BDHS: Bangladesh Demographic and Health Survey; MICS: Multiple Indicator Cluster Survey.

Fig 3 presents the absolute and relative socioeconomic inequalities in CISAF across wealth quintiles for each survey year. The SII values were consistently negative, indicating that CISAF was disproportionately concentrated among children from the poorest households in all three surveys. However, the inequality was highest in 2017–18 (Coef: -0.12; 95% CI: -0.14, -0.10), compared to 2019 (Coef: -0.10; 95% CI: -0.11, -0.08) or 2022 (Coef: -0.09; 95% CI: -0.12, -0.06) (Fig 3).

Similarly, RII values indicate that the log odds of experiencing CISAF remained higher among the poorest children compared to the richest. However, the odds of CISAF among poorest children was highest in 2019 (OR: 1.54; 95% CI: 1.46, 1.64) compared to 2017–18 (OR: 1.41; 95% CI: 1.29, 1.51) or 2022 (OR: 1.39; 95% CI: 1.25, 1.57) (Fig 3).

## Discussion

This study examined national trends and socioeconomic factors associated with the CISAF among children under five in Bangladesh using three nationally representative surveys, namely BDHS 2017–18, MICS 2019, and BDHS 2022. The CISAF offers a more holistic measure of severe undernutrition by encompassing multiple indicators, i.e., severe stunting, wasting, and underweight, in a single composite metric. A declining trend in CISAF prevalence was observed from 2017 to 2022, while persistent socioeconomic disparities continued to highlight the importance of equity-focused interventions.

The national prevalence of CISAF declined from 10.3% in 2017–18 to 7.6% in 2022. This decline is consistent with recent improvements reported in national nutrition indicators in Bangladesh, potentially driven by improvements in maternal education, antenatal care, public health outreach, and poverty reduction efforts [13–15]. Previous studies also reported improvements in child nutrition indicators such as stunting and underweight in Bangladesh over time [24]. However, the results reinforce the concern that traditional indicators may underestimate the burden of severe undernutrition when used independently, thus supporting the broader application of CISAF in nutritional surveillance [5].

Despite the decline, the prevalence remains notable, particularly given the serious developmental, cognitive, and intergenerational consequences associated with severe undernutrition [2]. Moreover, the proportion of children experiencing severe stunting alone remained the largest contributor to CISAF, albeit declining across the years. These findings echo prior regional observations that stunting is more prevalent than other forms of anthropometric failure and persists even in moderately improving contexts [3,25].

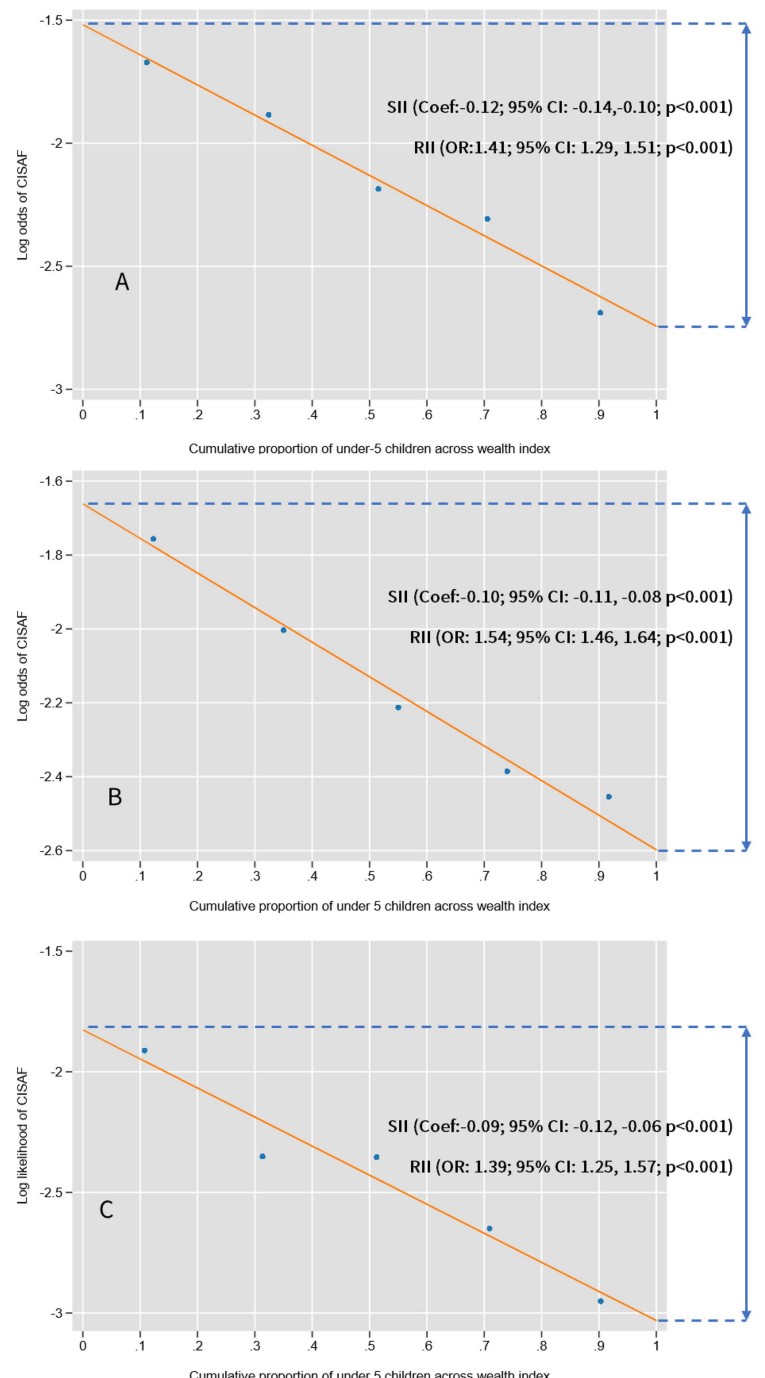

CISAF: Composite index of severe anthropometric failure; SII: Slope index of inequality; RII: Relative index of inequality; OR: Odds ratio; CI: Confidence interval; BDHS: Bangladesh Demographic and Health Survey; MICS: Multiple Indicator Cluster Survey

**Fig 3. SII and RII for the prevalence of CISAF according to wealth quintiles across survey years; (A) for BDHS 2017–18; (B) for MICS 2019; (C) BDHS for 2022.**

The study findings confirm that severe undernutrition is disproportionately concentrated among children from socio-economically disadvantaged households. Using SII and RII, higher levels of CISAF were observed among children from the poorest quintiles across all three survey rounds. The RII in 2019 was particularly elevated compared to 2017–18 and 2022, suggesting that inequalities may have slightly intensified during the middle survey period. These findings are consistent with global literature, indicating that relative gaps in child malnutrition have not narrowed uniformly across income groups, even in contexts where overall prevalence declines [3].

Wealth-related disparities may reflect differences in access to nutritious food, clean water, sanitation, and health services [26]. Lower-income households are more vulnerable to food insecurity and infectious diseases, both of which are key drivers of child undernutrition [8]. Moreover, social determinants such as housing conditions and maternal knowledge influence care practices and exposure to risk factors [5].

Maternal education emerged as one of the most significant predictors of CISAF across survey years. Children whose mothers had a secondary or higher level of education consistently had lower odds of CISAF. This aligns with earlier findings from multi-country analyses that maternal education is a strong and consistent determinant of a child's health and nutrition [3]. Educated mothers are more likely to adopt positive health behaviours, access healthcare services, and have higher nutritional awareness, all of which contribute to better child outcomes [27,28].

The analysis indicated that antenatal and prenatal care were associated with lower odds of severe undernutrition. Mothers with greater access to antenatal visits were less likely to have severely undernourished children [29], although this association varied in strength across the years. Media exposure, a proxy for health information access, was similarly associated with lower odds of CISAF, suggesting the positive impact of behaviour-change communication interventions on childhood malnutrition [29]. Interestingly, the effect of maternal media exposure and prenatal care did not show significant heterogeneity across survey years, implying consistent protective effects. This consistency supports prior recommendations that expanding mass media-based health communication and maternal services could yield sustainable improvements in child nutrition [9].

The decline in CISAF between 2019 and 2022 cannot be interpreted as evidence of improved service delivery during the COVID-19 pandemic. In Bangladesh, pandemic-related disruptions significantly affected routine maternal and child health services, with growth monitoring, vaccinations, and antenatal care experienced interruptions [30,31]. Concurrently, widespread economic shocks exacerbated food insecurity in the poorest families, and studies documented a substantial increase in rural household food insecurity during the lockdown period [31]. More detailed post-pandemic data will be necessary to disentangle these effects.

One of the main strengths of this study is the use of large, nationally representative datasets collected at different time points. This allows a strong analysis of both trends and inequalities over time. Using the CISAF also adds value, as it gives a more complete picture of severe undernutrition. Unlike traditional indicators, CISAF captures multiple forms of deprivation at once, making it a more sensitive tool. However, there are also some limitations to consider. First, the surveys were cross-sectional, which limits causal interpretation. Second, anthropometric data in BDHS 2022 were incomplete due to missing measurements, which may have introduced bias. Third, morbidity variables such as diarrhoea, fever, and cough were based on maternal recall, and thus subject to reporting error. Some potentially relevant determinants of undernutrition, such as dietary diversity, household food security etc., were not included in the analysis. The trend analysis covered a five-year period, which reflects recent national conditions but does not capture longer-term historical changes. Future work could examine sex-specific inequality patterns which was beyond the present study's scope. Although the datasets were harmonised carefully, small differences between DHS and MICS data collection procedures may remain, which could influence comparisons.

A clearer translation of these findings into programmatic action is essential for national planning. The observed socioeconomic gradients align with priority areas identified in the National Plan of Action for Nutrition [32]. Strengthening community nutrition workers, expanding maternal nutrition counselling, and ensuring the continuity of antenatal and postnatal

contacts may help address the persistent disadvantages faced by poorer households. Integrating CISAF into routine growth-monitoring platforms could support earlier identification of children at highest risk and improve referral efficiency. Social protection instruments, including cash transfers and food supplementation schemes, remain relevant for households in the lowest wealth quintiles, where the burden of severe anthropometric failure is most concentrated.

## Conclusion

The study demonstrates progress in reducing severe undernutrition among children under five in Bangladesh, as reflected in the declining prevalence of CISAF across survey years. However, persistent socioeconomic disparities remain, with children from the poorest households and those with less educated mothers disproportionately affected. These findings indicate that universal strategies alone are insufficient.

To address these gaps, nutrition programmes should adopt equity-focused approaches that prioritise disadvantaged groups. Targeted community-based interventions, conditional cash transfer schemes, and efforts to expand maternal education are particularly important. Integrating CISAF into national nutrition surveillance could strengthen policy responses by capturing multiple forms of undernutrition and identifying high-risk groups more effectively than single indicators. Strengthening community-level services such as growth monitoring, counselling on infant and young child feeding, and linking vulnerable families to social protection schemes will be essential to achieve more equitable and sustainable improvements in child nutrition.

## Supporting information

**S1 Fig. Categories for the extended composite index of anthropometric failure (CISAF).**
(TIF)

**S2 Fig. Components of CISAF across surveys.**
(TIF)

## Acknowledgments

Gratitude is extended to the interviewees and to all MICS survey staff involved in project planning and data collection. icddr,b is also grateful to the governments of Bangladesh and Canada for providing core/unrestricted support.

## Author contributions

**Conceptualization:** Md Fuad Al Fidah, Tasnuva Sarowar.

**Data curation:** Md Fuad Al Fidah, Tahia Tul Islam, Syeda Sumaiya Efa.

**Formal analysis:** Md Fuad Al Fidah, Md Nafis Fuad.

**Methodology:** Md Fuad Al Fidah, Syeda Sumaiya Efa.

**Writing – original draft:** Md Fuad Al Fidah, Md Nafis Fuad, Tahia Tul Islam.

**Writing – review & editing:** Tasnuva Sarowar, Syeda Sumaiya Efa.

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
