## [Decision Letter · Decision Letter 0]

29 Sep 2025

PGPH-D-25-02016

National Trends and Inequalities in Severe Anthropometric Failures in Bangladesh: Evidence from National Surveys

Dear Dr. Al Fidah,

Thank you for submitting your manuscript to PLOS Global Public Health. After careful consideration, we feel that it has merit but does not fully meet PLOS Global Public Health’s publication criteria as it currently stands. Therefore, we invite you to submit a revised version of the manuscript that addresses the points raised during the review process.

Please note that we have only been able to secure a single reviewer to assess your manuscript. We are issuing a decision on your manuscript at this point to prevent further delays in the evaluation of your manuscript. Please be aware that the editor who handles your revised manuscript might find it necessary to invite additional reviewers to assess this work once the revised manuscript is submitted.

Please carefully review the reviewer's comments and revise your manuscript accordingly. Please note that you are not required to cite the references suggested by the reviewer unless you feel they are relevant and would improve your introduction.

We look forward to receiving your revised manuscript.

Kind regards,

Sarah Jose, Ph.D.

Staff Editor

Journal Requirements:

Additional Editor Comments (if provided):

Reviewer #1:

Reviewers' comments:

Reviewer's Responses to Questions

**Comments to the Author**

1. Does this manuscript meet PLOS Global Public Health’s publication criteria?

Reviewer #1: Partly

2. Has the statistical analysis been performed appropriately and rigorously?

Reviewer #1: Yes

3. Have the authors made all data underlying the findings in their manuscript fully available (please refer to the Data Availability Statement at the start of the manuscript PDF file)?

Reviewer #1: Yes

4. Is the manuscript presented in an intelligible fashion and written in standard English?

Reviewer #1: No

Reviewer #1: The manuscript presents valuable insights into the trends and disparities in severe anthropometric failures among children under five in Bangladesh using nationally representative survey data. The strength of the study lies in its comprehensive assessment of the Composite Index of Severe Anthropometric Failure (CISAF) and its focus on socioeconomic inequalities. However, several areas require revision to enhance the manuscript's clarity and policy relevance.

Firstly, the introduction would benefit from a more compelling articulation of the global significance of childhood severe undernutrition and the rationale for adopting CISAF over traditional indicators. A clearer depiction of the study’s contribution to existing literature would strengthen the context. Cite the following studies:

https://doi.org/10.1016/j.nut.2021.111565

https://doi.org/10.1108/NFS-03-2023-0055

Secondly, the methodology section presently lacks detail regarding sample selection, variable definitions, and the statistical procedures employed. A more explicit description of the analytical methods, including how inequalities were quantified, is essential for reproducibility.

Furthermore, the paper should better extract and emphasize policy implications, such as targeted interventions for vulnerable groups identified by the study. A discussion of the limitations, including potential biases or unmeasured confounders, should be acknowledged to provide balanced conclusions.

Overall, addressing these points will significantly improve the manuscript's clarity, policy relevance, and scholarly rigor.

**Do you want your identity to be public for this peer review?** For information about this choice, including consent withdrawal, please see our Privacy Policy

Reviewer #1: **Yes:** Dr Om Raj Katoch

---

## [Decision Letter · Decision Letter 1]

28 Nov 2025

PGPH-D-25-02016R1

National Trends and Inequalities in Severe Anthropometric Failures in Bangladesh: Evidence from National Surveys

Dear Dr. Al Fidah,

Thank you for submitting your manuscript to PLOS Global Public Health. After careful consideration, we feel that it has merit but does not fully meet PLOS Global Public Health’s publication criteria as it currently stands. Therefore, we invite you to submit a revised version of the manuscript that addresses the points raised during the review process.

We look forward to receiving your revised manuscript.

Kind regards,

Helen Howard

Staff Editor

Journal Requirements:

Additional Editor Comments (if provided):

Reviewers' comments:

Reviewer's Responses to Questions

**Comments to the Author**

Reviewer #1: (No Response)

Reviewer #2: All comments have been addressed

Reviewer #3: (No Response)

publication criteria?

Reviewer #1: Yes

Reviewer #2: Yes

Reviewer #3: Partly

3. Has the statistical analysis been performed appropriately and rigorously?

Reviewer #1: Yes

Reviewer #2: Yes

Reviewer #3: Yes

4. Have the authors made all data underlying the findings in their manuscript fully available (please refer to the Data Availability Statement at the start of the manuscript PDF file)?

Reviewer #1: Yes

Reviewer #2: Yes

Reviewer #3: Yes

5. Is the manuscript presented in an intelligible fashion and written in standard English?

Reviewer #1: Yes

Reviewer #2: Yes

Reviewer #3: Yes

Reviewer #1: The authors have improved the manuscript by clarifying data presentation and strengthening methodological details. The analysis of trends and inequalities in severe anthropometric failure using national survey data offers valuable insights into child nutrition in Bangladesh. However, key issues remain. The manuscript lacks a clear discussion of policy implications and practical applications of the findings. Explicitly linking results to national nutrition policies or targeted interventions would enhance its relevance. Overall, while the manuscript has advanced, a more comprehensive and policy-oriented discussion is needed to increase its impact and usefulness for stakeholders addressing child undernutrition in Bangladesh.

Reviewer #2: The comments have adequately addressed and the manuscript is ready for publication.

Reviewer #3: Journal Name: PLOS Global Public Health

Manuscript Title: National Trends and Inequalities in Severe Anthropometric Failures in Bangladesh: Evidence from National Surveys

Manuscript Number: PGPH-D-25-02016R1

Thank you very much for giving me the opportunity to review the manuscript “National Trends and Inequalities in Severe Anthropometric Failures in Bangladesh: Evidence from National Surveys”. The relevance of this paper is notable, particularly in the context of Bangladesh. It utilises nationally representative data to explore nutritional disparities between boys and girls under the age of five. This research can increase scientific understanding. However, I recommend a thorough major revision to improve the manuscript’s presentation and overall impact.

REVIEWER COMMENTS

SECTION-WISE OBSERVATION

Title: This is partially acceptable

• It lacks relevance and does not adequately reflect the study's objective. I recommend revising the title to better align with the content of the research. Additionally, while the study focuses on CISAF, the mention of "Severe Anthropometric Failures" seems misplaced and should be clarified or adjusted accordingly.

Keywords: This is partially acceptable

• Please avoid repeating terms from the title in the keywords section.

• Acronyms such as "DHS," "MICS," "SII," and "RII" should be spelled out to enhance understanding, as they may not be familiar to all readers.

Abstract: Partially Acceptable; moderate corrections are required

• You have collected data from BDHS 2017-18, MICS 2019, and BDHS 2022, leading to a trend analysis covering a relatively short time span of only five years. For a more robust analysis, I recommend using data spanning at least a decade.

• Is this study intended as a longitudinal study? You note that “Higher odds of CISAF were associated with increasing age of the child (aOR: 1.03, 1.10 and 1.02, respectively).” suggesting that children become CISAF as they grow older. This warrants further clarification or write the sentence correctly.

• Regarding your statement, “However, lower odds were linked to wealthier families and higher education of the mother.” while maternal education is undoubtedly significant, it would enhance your analysis to include paternal education as well.

• I suggest clarifying and refining the conclusion for a more precise presentation of your findings.

Introduction: Acceptable, but major corrections are required

• In the sentence "To overcome this, Svedberg proposed the Composite Index of Anthropometric Failure (CIAF), which incorporates all forms of failure.[1]" the reference is cited as Anik et al. (2021). It seems more appropriate for this claim to be attributed to Svedberg. Could you please clarify this?

• In your discussion regarding CISAF, you mention "Later, Vollmer et al. (2017) introduced the...in resource-poor settings.[5]". However, the reference here is listed as Saif and Anwar (2023). Please review this and ensure that proper attribution is given.

• I recommend that you recheck all the references thoroughly, as the arrangement appears to be inconsistent. A systematic review of citations would enhance the manuscript’s credibility.

• It may be beneficial to include further references to support your study. This could help substantiate your claims and provide a broader context for your findings.

Objective: Partially Acceptable; minor corrections are required

• The objective should be revised to provide a clear statement that reflects the title of the study.

• The text states, "Using three recent nationally representative surveys..." However, there is no mention of the names of these surveys. It would be helpful to include this information to provide clarity and context for the reader.

Methods: Partially Acceptable; moderate corrections are required

• The definition of implausible values for Z-scores appears to deviate from standard ranges, instead of "Implausible values were defined as weight-for-height Z-score (WHZ) <-5 or >5 SD, height-for-age Z-score (HAZ) <-5 or >5 SD, and weight-for-age Z-score (WAZ) <-6 or >5 SD.". Please refer to the WHO Multicentre Growth Reference Study Group's guidelines: "WHO child growth standards: length/height-for-age, weight-for-age, weight-for-length and body mass index for age: methods and development. Geneva: World Health Organization, 2006." Specifically, implausible values should be based on accepted thresholds for weight-for-height Z-score (WHZ), height-for-age Z-score (HAZ), and weight-for-age Z-score (WAZ).

• I noticed inconsistencies in the terminology used to refer to the target population. It would be beneficial to adopt a single term throughout the manuscript, such as "under-five children," "U5C," or "under five," to enhance clarity.

• The focus of this study is on "Inequalities"; however, the sex of the child (male/female) is utilized as an independent variable. I suggest a more thorough exploration of the potential implications of this variable in the context of inequalities.

• Additionally, I recommend including a section labelled "Study variables" that distinctly outlines the "Outcome variable" and the "Independent variable." It would also be beneficial to provide "Operational definitions for the variables" separately to enhance understanding.

Results: Partially Acceptable, major corrections are required

• I recommend conducting the analysis separately for boys and girls, as the focus is on "inequalities." This approach may reveal factors that are specifically related to each gender, providing a deeper understanding of the issues at hand.

Discussion: Acceptable; minor corrections are required

• It may be beneficial to include further references to support your study.

Conclusion: This is acceptable

References: This is acceptable

OVERALL OBSERVATION

The suggestion for improving the language to avoid the use of "we" or "our" in the manuscript is well-founded. By opting for a more formal and objective tone, the authors can enhance the clarity and professionalism of their work.

**Do you want your identity to be public for this peer review?** For information about this choice, including consent withdrawal, please see our Privacy Policy

Reviewer #1: **Yes:** Dr Om Raj Katoch

Reviewer #2: No

Reviewer #3: No

---

## [Decision Letter · Decision Letter 2]

11 Jan 2026

National Trends and Inequalities in Severe Anthropometric Failures in Bangladesh: Evidence from National Surveys

PGPH-D-25-02016R2

Dear Dr Al Fidah,

We are pleased to inform you that your manuscript 'National Trends and Inequalities in Severe Anthropometric Failures in Bangladesh: Evidence from National Surveys' has been provisionally accepted for publication in PLOS Global Public Health.

Best regards,

Julia Robinson

Executive Editor

Reviewer Comments (if any, and for reference):

Reviewer's Responses to Questions

**Comments to the Author**

Reviewer #2: All comments have been addressed

publication criteria?

Reviewer #2: Yes

3. Has the statistical analysis been performed appropriately and rigorously?

Reviewer #2: Yes

4. Have the authors made all data underlying the findings in their manuscript fully available (please refer to the Data Availability Statement at the start of the manuscript PDF file)?

Reviewer #2: Yes

5. Is the manuscript presented in an intelligible fashion and written in standard English?

Reviewer #2: Yes

Reviewer #2: All comments have been satisfactorily addresses in my opinion. There are some areas that one reviewer requested additional data that authors did not have access to. I think it is very reasonable for the authors to push back on this request. While reviewer is correct that the request data would be interesting, the analysis the authors have presented is internally valid.

**Do you want your identity to be public for this peer review?** For information about this choice, including consent withdrawal, please see our Privacy Policy

Reviewer #2: No
